1        Comment on "Design study for an airborne $N_2O$ lidar" by Kiemle et al. (2024)

2        Joel F. Campbell[1], Bing Lin[1], Zhaoyan Liu[2]

[1]NASA Langley Research Center, Hampton, VA 23681
joel.f.campbell@nasa.gov
[2]NASA Ames Research Center, Moffett Field, CA 94035

In a recent publication (Kiemle et al., 2024) the following was stated:
"Another low-power option for IPDA is (modulated) continuous-wave (cw) laser operation instead
of emitting pulsed signals (e.g., Campbell et al., 2020). For measurements with a precision
requirement below 1 %, however, the length of the atmospheric column must be known to an
accuracy of better than 3 m, which is only practicable with short laser pulses in combination with
a sufficiently large detection bandwidth (Table 3; Ehret et al., 2008). Alternatively, a precision
range finder had to be added, which annihilates the cost benefit of cw lidar."
We reported doing those things with a CW lidar system in the referenced publication without the
aid of an external range finder so this statement is incorrect. The CW technique was first used in
radar many decades ago to do ranging, so there really is not an advantage or necessity to choose
one technology over another if ranging is the only consideration. In fact, one of the main
applications for CW lidar is ranging. The same physics applies to either technology. If a narrow
pulse is required for target discrimination purposes, this can also be achieved with CW using a
modulation with a wide bandwidth. Once the matched filter transform is performed, a narrow
width synthetic pulse can be achieved. A 3 m resolution CW ranging lidar would require a
modulation bandwidth of $\Delta f = \frac{c}{2\Delta r} = 50 MHz$, where c is the light speed and $\Delta r$ is the ranging
resolution. This is not difficult especially if optical communication hardware is utilized. Another
method would be to use frequency modulated CW (FMCW; Gao and Hui, 2012) or phase
modulated CW (PMCW; Zhi et al., 2025) and heterodyne detection. In fact, FMCW lidar is used
in the auto industry to detect near objects (Kim et al., 2020).
Regarding the point of a narrow pulse (or alternatively a narrow synthetic CW pulse) being
required to do ranging down to 3 meters: If the field is cluttered by clouds or other features, we
would say this statement is probably true depending on the situation, but in clear sky conditions
where the ground is the only return, interpolation can be used. Interpolating lower resolution lidars
is not a particularly controversial technique and has been used extensively in the past (Hu et al.,
2007; Ai et al., 2011; Dobler et al., 2013; Lu et al., 2014; Campbell et al., 2014). If the field is
cluttered by closely spaced scatterers, one return could interfere with another to distort the shape
of the pulse and affect the range measurement. If the ground is the only return this is not likely to
occur except through ground topography, which would also affect a pulse lidar. In some pulse
lidars where there is variability from pulse to pulse, interpolation can be more problematic
depending on the system. However, CW does not suffer from this. Each modulation frame is
generated from a preset waveform and clock and there is a very high degree of repeatability. Not
only that, each synthetic pulse is generated from multiple sweeps in our processing, and the
interpolation is a natural feature of the way that we do the matched filtering, using a type of circular
Fourier interpolation by collapsing the Kronecker comb of the matched filter in the frequency
domain, so the results of the modified matched filter produces an interpolated synthetic pulse with
very good results (Campbell et al., 2014). As long as the signal is Nyquist sampled, the original
continuous signal can be recovered to within the limits of noise. This is the basic tenet of the
Nyquist-Shannon sampling theorem. Fourier interpolation (or at least our version of it designed
for circular correlations) is the most natural and accurate interpolation method for this type of band
limited signal.
The paper in question shows results from the multifunctional fiber laser lidar (MFLL), an
instrument that was developed over many years by Harris Corporation (now L3Harris) in
collaboration with NASA Langley Research Center. MFLL evolved from an instrument that used
a single frequency modulation for each wavelength that were orthogonal to one another in its early
development (Dobbs et al., 2008) to what it was on the Atmospheric Carbon and Transport –
America (ACT America) flights where it used orthogonal swept frequency modulations (Dobler et
al., 2013, Campbell et al., 2020). Although it is true ranging is more problematic with single
frequency modulations, swept frequency modulation is a technique commonly used in many older
CW radars to do ranging and that is what was being used by MFLL for the ACT America flights.
The results show that the ranging is clearly less than 3 m, which meets and, actually, exceeds the
science/instrumentation requirement. The authors have also experimented with PN code
modulation on different instruments with good results (Campbell et al., 2014).

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
