# Peer review of "Comment on "Design study for an airborne N2O lidar" by Kiemle et al. (2024) Joel F. Campbell1, Bing Lin1, Zhaoyan Liu2 2 3 1NASA Langley Research Center, Hampton, VA 23681 4 5 joel.f.campbell@nasa.gov 6 2NASA Ames Research Center, M"

_EGUsphere, 2025_

## Author Response (AR1)

Reviewer 1

"This manuscript/commentary presents valid, scientifically sound counterarguments to the quoted statement from Kiemle et al. 2024. However, it does so with a slightly defensive tone. While understandable, the defensive tone should be carefully removed so as to avoid being perceived as antagonistic and detracting from the merits of the arguments being made."

This seems like a reasonable request, but we will say many of us were surprised and bewildered why he would make such comments on a paper describing a major campaign without any kind of analysis or proper explanation for dismissing our results. Not having to respond to anything like this before we weren't sure exactly how to do it or what the tone should be. Our original plan was to request a retraction, but we were asked to respond this way instead. Taken at face value it would seem he was saying we couldn't have done what we said we did because only pulse lidar can measure range to the required accuracy without an external lidar separate to the CW system. The implication is that we either did not know what we were doing or falsified our results. Maybe he did not mean it that way but that is the way it reads. The author and coauthors of this response are not aligned to any technology. Our involvement was more in the analysis part of it. Had the comments not addressed us personally we would have ignored them. The first author's involvement in this started around 2010 when he was part of the ASCENDS program. At that time his involvement was peripheral, and he was working independently on PN code modulation, which he found superior because of the superior autocorrelation properties. By the time of the Act America flights, he was more of a central figure as the data manager, algorithm development lead, and the person who developed all the software for signal processing and retrievals. He was particularly surprised because he met Kiemle before at the EGU where he had a poster presentation in 2014 where he was presenting his PN code work. He explained to Kiemle in detail how CW worked and how one gets a synthetic pulse from a CW system, that in this way it was very similar to pulse lidar and the same physics applied. We have also seen that team elsewhere at the AGU where these concepts were presented by us and Goddard.

"The statement by Kiemle regarding the use of an additional precision range finder seems to be either an innocent misunderstanding or an attempt at misdirection, as Campbell et al. 2020 makes no mention of such a measurement. To the contrary, Campbell describes the use of a fine interpolation technique to achieve sub-meter ranging accuracy from the swept waveform used to make the XCO2 measurement (with references for further detail). While it is true that previous versions of the MFLL instrument have used a dedicated pseudo-random noise (PN) altimeter, that was not the case for the ACT-America campaign, and the Campbell 2020 paper in question makes no mention of a PN altimeter."

It's interesting you should mention this PN altimeter because that is also a CW instrument with 1 meter resolution, which is also counter to what Kiemle claims – that a pulse lidar is required to achieve the required range accuracy. We want to be perfectly honest and state this altimeter flew on every MFLL flight we know of, but the only time it was used in retrievals was when they

were using single frequency per channel modulations in the time frame before 2010. By the time the first author was involved around 2010, they had already started using swept frequency modulation. By around 2012 the first author derived the autocorrelation function for their swept frequency modulation, and they were using that as a fit to their matched filter and used that for interpolation. In (Dobler et al., 2013) they compared those results with their PN altimeter, and they agreed to less than 3 m rms. When the Act America campaign started around 2015, we converted all the processing the first author developed for PN codes including the hyperfine interpolation technique and adopted it for swept frequency. At that time the PN altimeter was available but not used for anything other than a sanity check. Harris/L3Harris was working independently, and they were still using the techniques outlined in (Dobler et al., 2013) with good results, and they have recently confirmed they also did not use the PN altimeter either for retrievals even though it was available to them.

"I suggest the authors of this commentary clean up the text to be a bit more formal.  Suggestions include:

- Consider rewording lines 15-16 to be less defensive (e.g., "at a loss to understand what is being implied here")"

Changed to "We reported doing those things with a CW lidar system in the referenced publication without the aid of an external range finder, so this statement is incorrect."

- "Rewrite lines 28-29 to be less conversational and more in the style of formal technical writing"

Done

"Technical corrections:

- Eliminate contractions (lines 17, 54)"

Done

- "Capitalize CW initialism, as was done with PMCW and FMCW"

Done

- "The sentence in lines 36-37 that begins, "In some pulse lidars…" is not a complete sentence."

Done.

- "Harris Semiconductor has not existed since 1999.  At the time of the ACT-America program, it was Harris Corporation.  It is now L3Harris Technologies."

Done.

Reviewer 2

"I have reviewed this comment on the Kiel et al. 2024 paper. Its essential point is that in principal modulated CW lidar may also be used to remotely measure range with a resolution of < 3m. So its authors have a valid point that a pulsed lidar approach is not actually required for this, as stated in the Kiemle paper.

However I recommend that the comment be shortened to emphasize this point, and to reduce the large number of self references by its authors. I also recommend the phrase "CW lidar" be replaced with "modulated CW lidar," since the output of the CW laser has to be modulated to allow measuring range."

Unless there is a strong objection, we would preferer to keep it the length as it is now. The reason being is we feel it is necessary to justify the various points being made and our response is not excessively long. Not counting the quote from Kiemle's article there are 3 paragraphs and Kiemle makes 3 claims that need to be addressed. The first is that CW can't be used for ranging (or at least high resolution ranging as he is not clear on this point). Paragraph 1 addresses this issue. The second claim is that a narrow pulse from a pulse lidar is required to measure range to the required accuracy. Paragraph 2 addresses the narrowness of the pulse issue and how we achieve the required accuracy using interpolation in clear sky conditions. The third claim is that an external altimeter is required with a CW lidar system. Paragraph 3 addresses this issue and goes through the history of this instrument and why there might have been a confusion based on an early version of it when they used single frequency orthogonal modulation. In its first iteration in before 2010, it actually did require an external altimeter but all that changed after 2010 when they started using swept frequency modulation. This is addressed in the 3rd paragraph.

Regarding changing this to modulated CW comment, Kiemle seems to be fully aware of this since he himself uses the term "(modulated) cw". Although it is true it must be modulated in a particular way to be useful for ranging, it is equally true that not all modulated CW is useful for ranging. The prime example of this is in the early phase of this instrument where they used single frequency modulation. In order to be useful for differential absorption there must be a way to separate on and offline wavelengths. In the original early development, they were using single frequency orthogonal modulations. That way the two wavelengths could be separated with a software based "lock-in" detection as they called it. However, this type of modulation is near useless for determining range, which is why they later chose an orthogonal swept frequency modulation. The orthogonality allows one to separate the on and offline wavelengths while the swept frequency part of it allows them to do ranging at the same time. It was quite an innovative technique that Harris/L3 Harris developed. In the paper we wrote we called it IM-CW and the title used the term "Intensity-Modulated Continuous-Wave Differential Absorption Lidar" so there never was any confusion of the type you mention except relating to this response which Kiemle himself has not addressed at all.

Regarding the self cites issue, this is a bit confusing because we only use 2 from the lead author in the reference list. One is the article in question and the other explains the interpolation used which we think is necessary to show our methods and address Kiemle's comments. It is true these two cites are addressed multiple times in the response but there are only two citations and they are necessary to make the points being made.